# *Encephalitozoon hellem* Infection Promotes Monocytes Extravasation

**DOI:** 10.3390/pathogens11080914

**Published:** 2022-08-15

**Authors:** Yishan Lu, Guozhen An, Xue Wang, Yunlin Tang, Jiangyan Jin, Jialing Bao, Zeyang Zhou

**Affiliations:** 1Sate Key Laboratory of Silkworm Genome Biology, Southwest University, Chongqing 400715, China; 2Chongqing Key Laboratory of Microsporidia Infection and Control, Southwest University, Chongqing 400715, China; 3College of Life Sciences, Chongqing Normal University, Chongqing 400038, China

**Keywords:** *Encephalitozoon hellem*, monocytes, extravasation, peripheral blood, bone marrow

## Abstract

**Background:** Microsporidia are a group of obligated intracellular fungus pathogens. Monocytes and the derivative macrophages are among the most important players in host immunity. The invasion of microsporidia may significantly affect the monocytes maturation and extravasation processes. **Methods:** We utilized a previously established microsporidia infection murine model to investigate the influences of microsporidia *Encephalitozoon hellem* (*E. hellem*) infection on monocyte maturation, releasing into the circulation and extravasation to the inflammation site. Flow cytometry and qPCR analysis were used to compare the monocytes and derivative macrophages isolated from bone marrow, peripheral blood and tissues of *E. hellem*-infected and control mice. **Results:** The results showed that the pro-inflammatory group of CD11b^+^Ly-6C^+^ monocytes are promoted in *E. hellem*-infected mice. Interestingly, the percentage of Ly-6C^+^ monocytes from *E. hellem*-infected mice are significantly lower in peripheral blood while significantly higher in the inflamed small intestine, together with up-regulated ratio of F4/80 macrophage in small intestine as well. **Conclusions:** Our findings demonstrated that *E. hellem* infection leads to promoted monocytes maturation in bone marrow, up-regulation of extravasation from peripheral blood to inflammation site and maturation into macrophages. Our study is the first systematic analysis of monocytes maturation and trafficking during microsporidia infection, and will provide better understanding of the pathogen–host interactions.

## 1. Introduction

*Encephalitozoon hellem* belongs to microsporidia which is defined as a huge group of eukaryotic, intracellular pathogens. They can infect a variety of hosts including human, specifically the immunocompromised populations [1]. The common infection sites include host’s digestive tract and the symptoms varies from latent infection to systemic inflammation and even death of hosts [2,3,4]. Since the first report of human microsporidian infections was been known in 1959 [5], the research on microsporidiosis has attracted much more attention.

Now we know that the host innate immunity plays a significant role against microsporidia, such as the interactions between microsporidia and macrophages [6,7,8]. As key regulators of innate immunity and inflammation, monocytes and macrophages mainly play a role in pathogen clearance, antigen presenting, induction of adaptive immune response and so on [9]. It has been reported in relevant reports that localized macrophages, especially in the digestive tract, rapidly recognize the microsporidia through the variety of receptors containing the pattern recognition receptor [10] and result in producing numerous defense mediators, including cytokines and reactive nitrogen and oxygen species [11]. However, there is no knowledge about how microsporidia infection affects macrophages derived from precursor monocytes.

Monocytes and the derivative macrophages, regarded as the immediate arm of the immune system, have the essential function of effecting and regulating the innate immune response [11,12]. Monocytes develop from the progenitors in bone marrow, trafficking through blood to peripheral tissue and existing as precursors of macrophages and DCs. Cell populations are usually categorized by their expressed cell surface markers. The CD11b is a general marker for monocytes, monocyte derivatives and some NKs as well, while the Ly-6C^+^ is usually considered as specific marker for “inflammatory” monocytes [13,14,15,16]. To gain the ability of clearance and control of fungal, protozoal and other pathogenic infection, the recruitment of monocytes is vital, especially Ly-6C^+^ monocytes. Previous research has shown that, at the post-infection stage of *Encephalitozoon cuniculi* or *Encephalitozoon intestinalis*, CC-chemokines of CCL2, CCL7, et al., were upregulated [13]. Both CCL2 and CCL7 play a role in binding to CCR2 to mediate the egress of Ly-6C^+^ monocytes which further traffic from bone marrow to the blood circulation and derive to inflammatory macrophages with high expression of F4/80 [17]. During migration, peripheral monocytes extravasate from blood vessels and enter inflamed tissues, and this process is often affected by pathogens [18,19]. For instance, in the period of acute inflammation of intracellular bacteria *L. monocytogenes*, blood monocytes decrease during this time, as reported previously [20].

To fill the gap in knowledge of how *E. hellem* infection affects monocytes maturation and extravasation, we used the established murine model [21]. Monocytes were isolated from *E. hellem*-infected and uninfected control mice, flow cytometry and qPCR analysis were applied. Our study will help us to better understand the long-term influences of microsporidia infection on host immunity establishment.

## 2. Materials and Methods

### 2.1. Mice

Specified pathogen-free, 6-week-old female C57BL/6 mice were purchased and reared in animal care facility. All of the experiment mice were conducted following the Southwest University-approved animal protocol (SYXK-2017-0019).

### 2.2. Pathogen

*Encephalitozoon hellem* strain (ATCC 50504/50451), gifted by Prof. Louis Weiss (Albert Einstein College of Medicine, New York, NY, USA), was used in this study. The spores were reproduced in rabbit kidney cells (RK13, ATCC CCL-37) supplied with 5% CO_2_ in Minimum Essential Medium Eagle (MEM) containing 10% fetal bovine serum (FBS) (ThermoFisher, Waltham, MA, USA) and penicillin-streptomycin. The spores were collected from culture media and then passed through 5 µm-size filter (Millipore, Billerica, MA, USA) to remove host cells, then concentrated by centrifugation and stored in sterile distilled water at 4 °C [22].

### 2.3. Experimental Design

This study used a previously established mouse infection model, which uses dexamethasone to transiently suppress mice immunity for a short period of time [21]. Dexamethasone (Aladin, Cas 2392-39-4, Shanghai, China) was injected to intraperitoneal at the dose of 5 mg/kg for each mouse on a daily basis, which persisted for six days [23].

The mice were then intraperitoneally injected with 1 × 10^7^
*E. hellem* spores per day for two days. Controls were uninfected mice. All the mice sacrificed on the 17th day after infection, and samples of bone marrow (from hind legs), peripheral blood, about 0.32 g of small intestine were collected from each group. TRIzol method was used to extract the total RNA of samples. CTAB method was used for the extraction of the whole genome in tissues. PCR assay was used to prove the infection.

### 2.4. Bone Marrow Isolation

Sterile scissors were used to open the abdominal cavity, then removed the surface muscles to find the pelvic-hip joint. A sterile blade and tweezers were used to remove the muscles, adipose tissues and other surrounding tissues. Then, the knee joint was cut off, and the operation was carried out in the plate containing PBS [24]. Both ends of the femur and tibia were cut off, and 10 mL cell dissociation buffer was used to flush the bone marrow into the plate. The bone marrow suspension was transferred to pass through the 70 μm nylon cell strainer using a 2.5 mL syringe into a 50 mL centrifuge tube for collecting filtered cells. The immune cells were collected and resuspended in FACs buffer (1× PBS + 1% FBS) after the lysis of erythrocyte through RBC lysis buffer.

### 2.5. Flow Cytometry Analysis

The isolated bone marrow single cells were used for the detection of surface marker expressions, CD11b and Ly-6C, by flow cytometry. 200 μL immune cell resuspension was taken out of each sample and filtered by cell strainers. The cell resuspension was then stained for 30 min at 4 °C by APC-conjugated anti-CD11b and PE-conjugated anti-Ly-6C (BioLegend, San Diego, CA, USA). About 10^6^ cells were recorded. Through the support of the laboratory, flow cytometry analysis could be performed by Beckman Coulter CytoFlex S, and the data were analyzed by CytExpert Software.

### 2.6. Peripheral Blood Collection

Peripheral blood samples were collected from mice in both groups—the *E. hellem*-infected and the uninfected control. Sodium citrate was used as anticoagulant, and blood samples were centrifuged at 400× *g* for 10 min to enrich plasma. All samples were then subjected to qPCR analysis.

### 2.7. Detection of E. hellem

The colonization of *E. hellem* in the host tissues was proved through PCR assay, using primers 5′-CACCAGGTTGATTCTGCCTGACGT-3′ and 5′-CCTCTCCGGAACCAAACCCTGAT-3′, 279 bp of *E. hellem* gene fragment can be amplified.

### 2.8. Reverse Transcription-Quantitative Polymerase Chain Reaction

By the method of TRIzol, total RNA from mice bone marrow cells, peripheral blood cells and intestines were extracted. Reverse transcription was conducted by using of Hifair^®^ 1st Strand cDNA Synthesis SuperMix for qPCR (Yeasen Biotechnology, Shanghai, China). qPCR was performed through Roche LightCycler^®^ 96 System (Roche, Shanghai, China) with the use of Hieff^®^ qPCR SYBR^®^ Green Master Mix (Yeasen Biotechnology, Shanghai, China). Actin was used as the internal reference in the whole investigation. Primers were either synthesized by using the Primer Premier 5.0 program or supported by previous studies. The sequences were shown as follows: 

CD11b, 5′-CAATAGCCAGCCTCAGTGC-3′ (forward) and 5′-GAGCCCAGGGGAGAAGTG-3′ (reverse);

Ly-6C, 5′-GCAGTGCTACGAGTGCTATGG-3′ (forward) and 5′-ACTGACGGGTCTTTAGTTTCCTT-3′ (reverse);

F4/80, 5′-ACCTAGACATCGAAAGCAAA-3′ (forward) and 5′-TGATTATGAAACAGCCAACA-3′ (reverse);

CCR2, 5′-GAGTGAGAAGGAGGAGATATGC-3′ (forward) and 5′-AACACAGATAGGAGAAGGAACC 3′ (reverse);

Actin, 5′-GCTGTCCCTGTATGCCTCTG-3′ (forward) and 3′-TGATGTCACGCACGATTTCC-5′ (reverse).

### 2.9. Statistical Analysis

For statistical analysis and groups comparison, one-way ANOVA and Student’s *t*-tests were implemented to figure out the difference. *p* < 0.5 was deemed to be the indication of significant difference between two groups.

## 3. Results

### 3.1. Effects of E. hellem Infection on Monocyte Population in Bone Marrow

Firstly, to prove the infection of *E. hellem* in the host tissue, a PCR assay was used. The result showed that *E. hellem* successfully infected the mice (Figure 1). Then, we detected the expressions of CD11b and Ly-6C by bone marrow cells at the transcription level by qPCR. The results showed the upregulation of both markers, yet no statistical significance at this level (Figure 2A,B).

We further investigated the expressions of CD11b and Ly-6C on bone marrow cells by flow cytometry analysis. We firstly assigned the monocytes as low granule content and high CD11b-expression cells among all bone marrow cells (Figure 3A). The expressions of both CD11b and Ly-6C on monocytes from *E. hellem*-infected mice were significantly higher than on the monocytes from uninfected controls (Figure 3B,C). These results indicated that CD11b^+^Ly-6C^+^ monocytes (the inflammatory monocytes) are promoted to get proliferation in the bone marrow under the circumstances of *E. hellem* infection.

### 3.2. Effects of E. hellem Infection on Monocyte Population in Peripheral Blood

After monocytes were produced in bone marrow, they will be released into peripheral for exerting their future functions. As the result, we next assessed and compared the monocytes populations in blood from *E. hellem*-infected and uninfected groups. qPCR was applied for the detection of the expression level of CD11b and Ly-6C in murine peripheral blood. The result indicated that although the expression of general marker of monocytes and other leukocytes showed significant upregulation (Figure 4A), interestingly, the expression of specific marker Ly-6C for monocytes was significantly downregulated in the contrast with the uninfected group (Figure 4B).

As the result, we infer that Ly-6C^+^ monocytes are either stocked in the bone marrow or exuding blood vessels in a large number to the inflamed tissues during the period of infection.

### 3.3. Monocytes and Derivatives Propagation in Small Intestine

Since microsporidia predominantly infect intestine and to further confirm our hypothesis that the monocytes extravasation in peripheral blood were upregulated after *E. hellem* infection, we next analyzed the monocytes and derivative populations in small-intestine samples. The qPCR analysis results showed that the expression levels of CD11b and Ly-6C were upregulated in the inflamed tissue of infected group (Figure 5A,B) and the marker of macrophages-maturating F4/80 was significantly upregulated (Figure 5C) compared to the controls, indicating up-regulated differentiation into derivative macrophages in the small intestine.

### 3.4. CCR2 Mediates Monocytes Egression and Extravasation

CC chemokine receptor 2 (CCR2) is necessary for monocytes egression from bone marrow and recruitment from the blood to inflamed tissue. The expression of CCR2 was detected in bone marrow, peripheral blood and small intestine through relative quantification. We found that the expression level of CCR2 shows upregulation in bone marrow (Figure 6A), but downregulation in peripheral blood and then upregulation in small intestine (Figure 6B,C). The expression pattern of CCR2 is in accordance with the expression patterns of Ly-6C in bone marrow, blood and inflamed tissue.

Taken together, we believe that the extravasation of monocytes from peripheral blood ascends, which promotes the extravasation of monocytes to the inflamed tissue so as to undergo further differentiation and function the clearance of pathogens.

## 4. Discussion

In this study, we found that the infection of *E. hellem* stimulates the response of inflammatory monocytes, promotes proliferation of this set of cells in bone marrow, and ascends the extravasation of monocytes in the peripheral blood. Our findings also demonstrate the *E. hellem* infection would promote these monocytes differentiated into residential macrophage in the inflamed tissue.

During infection, the Ly-6C^+^ monocytes group are recruited to the peripheral tissue through blood circulation, and then further differentiated into activated macrophages. The derivative macrophages then express cell surface marker F4/80 [25]. In steady-state or no-infection conditions, Ly-6C^+^ monocytes would gradually downregulate the expression of Ly-6C [26], while under the circumstance of infection, Ly-6C^+^ monocytes act as inflammatory monocytes with the high capacity to migrate to the site of inflammation [20]. Our findings of Ly-6C expression patterns of high expressions in bone marrow and inflammation tissue, which are consistent with these previous reports. In addition, the short-chain fatty acids produced by gut microbiota may also aid the monocyte maturation and egression. It is known that *Firmicutes*, one of the major producers of butyrate, cooperates with the regeneration of bone marrow and promotes the differentiation of hematopoietic stem cells by providing the irons and metabolites [27]. *E. hellem* infection can change the abundance of *Firmicutes* in microbiota [21]. According to previous studies, we can reasonably infer that the abundant change in gut microbiota and the metabolites after infection could promote the differentiation of hematopoietic stem cells into monocytes in the bone marrow.

In the peripheral blood, we found the expression level of Ly-6C decreased after *E. hellem* infection. It could be the result of the decreasing of absolute number of monocytes due to increased extravasation. This inference may be supported by the study that concluded *Leishmania major* infection leads to the activation of bone marrow HSPCs and enhances myelopoiesis in the bone marrow, with a decreased absolute number of monocytes in the peripheral blood [20,28]. In the acute inflammatory phase of *Listeria monocytogenes* infection, monocytic left shift occurs in the bone marrow at about 72 h [29]. Such a left shift was also reported in the chronic infection of *E. hellem,* which needs further exploration.

The main driving force of monocyte egression and extravasation is the chemokine CCLs, such as CCL7 interacting with the CCR2 expressed on the cell surface [30]. It is reported that the infection of microsporidia *Encephalitozoon cuniculi* up-regulated expression of CCL1, CCL2, CCL3, CCL7 and other chemokines [13]. In this study, we demonstrated the expression levels of CCR2 in bone marrow, peripheral blood and small intestine are significantly affected by *E. hellem* infection and are consistent with the expression change patterns of Ly-6C on monocytes after infection. All indicate that, under the condition of *E. hellem* infection, bone marrow monocytes are prompted to proliferate and egress to blood driven by CCLs-CCR interaction, and this chemotaxis force drives more monocytes extravasation into inflamed tissues.

The gastrointestinal tract is the major colonization site for microsporidia infection [3]. Therefore, measuring the expression levels of F4/80 by immune cells in this tissue would reflect the effect of macrophages in response to *E. hellem* infection. In this study, we confirmed the upregulated expressions of F4/80 in small intestine of *E. hellem*-infected mice. The up-regulation of differentiated macrophage and the CCR2 together indicate that microsporidia infection stimulates more macrophages to this immune response, yet maybe later or during long-term latent infection, these immune cells would be hijacked to act as a “Trojan horse”, which had been found under the infection of *Encephalitozoon cuniculi* [10,31].

More interestingly, the maturation and migration of dendritic cells are reported to be inhibited by microsporidia infection [32]. Therefore, we are very interested in future studies to investigate in depth that how microsporidia infection influences dendritic cells and some other kinds of innate immune cells.

## 5. Conclusions

In this study, we found that during the infection course of *E. hellem*, host monocytes in the bone marrow proliferated in response to the infection. The extravasation of monocytes also increased significantly upon reaching the site of infection through peripheral blood. In this way, the differentiation of monocytes to phagocytic cells such as macrophages were promoted. Our study is the first investigation in a systemic view about monocytes proliferation and extravasation in responses to *E. hellem* infection.

## Figures and Tables

**Figure 1 pathogens-11-00914-f001:**
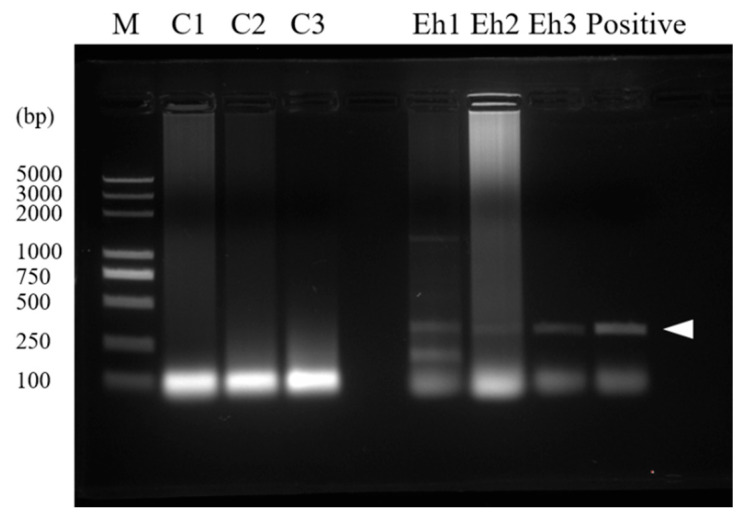
Confirmation of infection. Intestines were collected from *E. hellem*-infected and uninfected mice and the tissues were dissected; whole DNA was isolated and used as a template for PCR assay. Image of PCR result showed that the *E. hellem*-infected group had positive bands (as pointed out by arrow, 279 bp), indicating the successful infection of *E. hellem*.

**Figure 2 pathogens-11-00914-f002:**
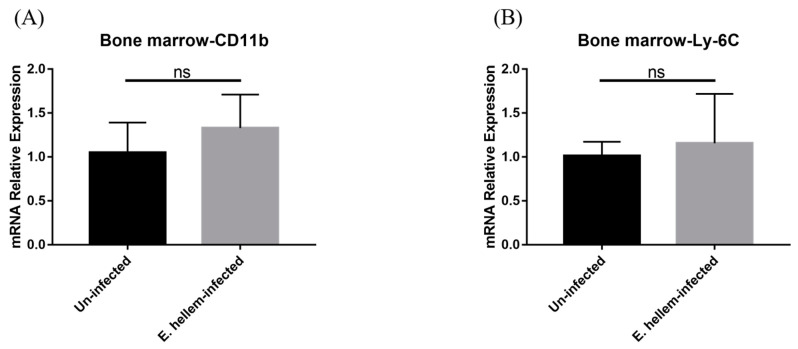
Cell markers expression. Bone marrow cells were collected and subjected to qPCR analysis. (**A**) The expression level of CD11b were upregulated in the *E. hellem*-infected group without significance. (**B**) The expression level of Ly-6C showed upregulation compared to uninfected mice. (ns = No Significance, *n* = 6 for *E. hellem*-infected group, *n* = 5 for uninfected control).

**Figure 3 pathogens-11-00914-f003:**
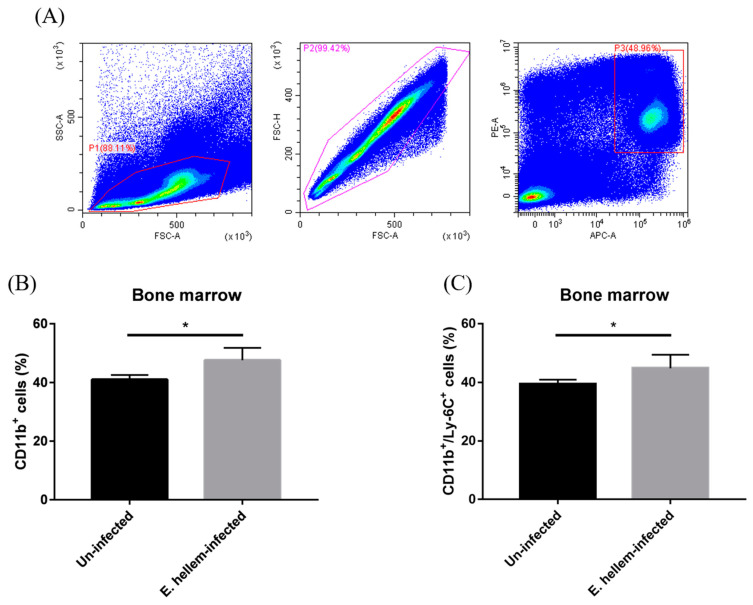
Flow cytometry analysis of monocytes. Bone marrow cells were isolated from *E. hellem*-infected and uninfected mice, respectively, the part of samples was subjected to flow cytometry analysis. (**A**) Bone marrow monocytes were labeled with CD11b Ab and Ly-6C Ab. Based on the characteristics of monocytes and removing of cell adhesion through FSC-A and FSC-H, gated cells (P1 + P2 + P3) were separated by cell sorting. (**B**) The ratio of SSC^low^CD11b^+^ monocytes showed the significant upregulation in the group of *E. hellem*-infected mice. (**C**) SSC^low^CD11b^+^Ly-6C^+^ cells refer to the inflammatory monocytes which were significantly upregulated compared to the uninfected group. (* = *p* < 0.05, *n* = 6 for *E. hellem*-infected group, *n* = 5 for uninfected group).

**Figure 4 pathogens-11-00914-f004:**
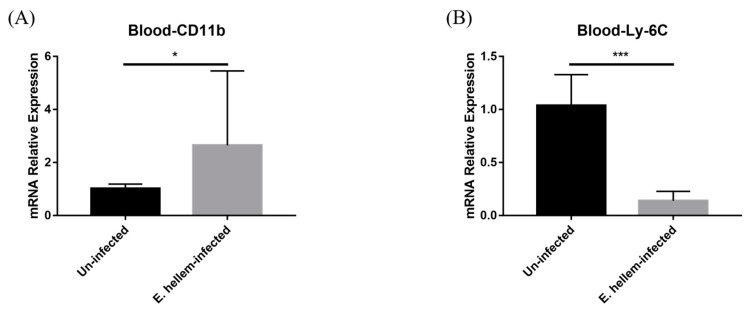
Cell markers expressions. Peripheral blood was collected from *E. hellem*-infected mice and uninfected mice, respectively, then subjected to qPCR analysis. (**A**) The expression level of CD11b, showing significantly upregulation compared with the uninfected group. (**B**) The expression of Ly-6C significantly decreased at the post-infection state. (* = *p* < 0.05, *** = *p* < 0.001, *n* = 6 for *E. hellem*-infected group, *n* = 5 for uninfected group).

**Figure 5 pathogens-11-00914-f005:**
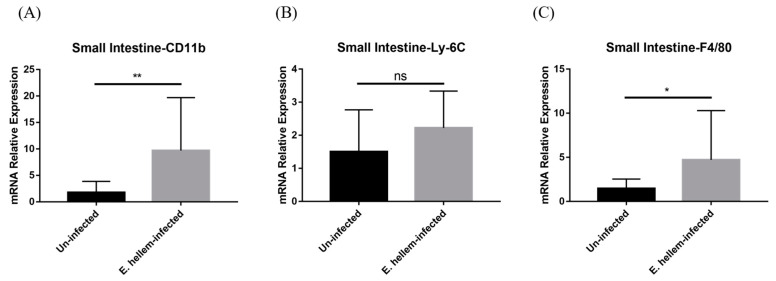
Cell surface markers expressions. The expression levels of CD11b, Ly-6C, F4/80 in small intestine were measured by qPCR in *E. hellem*-infected group and uninfected group, respectively. (**A**), the expression level of CD11b showed significant upregulation compared with uninfected group. (**B**), the expression of Ly-6C ascended without significance between two groups. (**C**), the expression of F4/80 was upregulated significantly after infection of *E. hellem*. (* = *p* < 0.05, ** = *p* < 0.01, ns = No Significance, *n* = 6 for *E. hellem*-infected group, *n* = 5 for uninfected group).

**Figure 6 pathogens-11-00914-f006:**
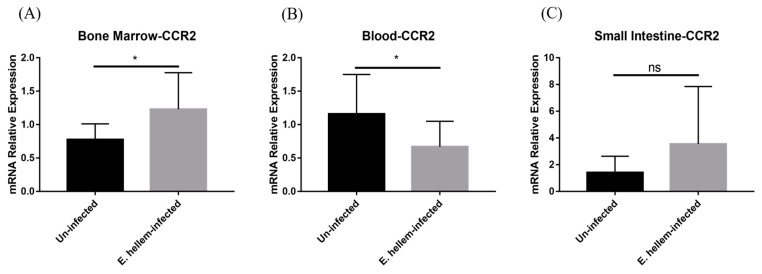
Chemokine receptor expression. The expression level of CCR2 was detected in bone marrow, peripheral blood and small intestine from *E. hellem*-infected mice and uninfected mice, respectively, by means of qPCR. (**A**) The expression of CCR2 in bone marrow, upregulated with significance compared to the controls. (**B**) The expression of CCR2 was significantly downregulated in peripheral blood after infection. (**C**) CCR2 expression in small intestine was upregulated after *E. hellem*-infecting. (* = *p* < 0.05, ns = No Significance, *n* = 6 for *E. hellem*-infected group, *n* = 5 for uninfected group).

## Data Availability

The original data and materials are available upon request.

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
