# Peer review of "Encephalitozoon hellem Infection Promotes Monocytes Extravasation"

_pathogens, 2022, doi:10.3390/pathogens11080914_

Round 1
Reviewer 1 Report
Considering that the immune response is the mainstay for the fight against microsporidiosis, the findings of this paper are interesting to contribute to the clarification of the assembly of the immune response against these agents, however, some modifications should be made to improve the understanding of the design, the results and concussion.
Introduction: Most of the cited papers are more than 10 years old, we suggest updating these references and focusing more on the role of phagocytic cells in microsporidiosis.
Material and methods:
Some concerns involve the experimental model and should be clarified:
- What is the inoculation route of E. hellem spores used for the infection?
- How was the infection diagnosed? Considering that not all animals become infected, as described in the base article for this model, perhaps some animals included in the infected group could not be infected? How to prove encephalitozoonosis?
- How many days after infection were the collections performed?
- Why use the intestine to identify monocytes?
- Which tissues should be most affected by the infection?
- What is the dose of dexamethasone used? It wasn't clear.
To be more precise, monocyte and phagocyte populations should be identified in the gut, as done in the bone marrow. When inferring an inflammatory process in the intestine, it must be identified either by counting cells by cytometry or by histopathological analysis. Which infestinal segment was used for extraction? what is the extraction method? what weight of intestine did you use? The relative expression was not clarified! Considering that the expression refers to the number of cells is imprecise, since few cells can express many of the investigated molecules, does not seem to be a sufficient approach to answer the proposed questions.
Results:
- It is suggested that figures 1 and 2 are unified.
- The text referring to figure 4 suggests an increase in Ly6C in the intestine of infected mice, but the figure shows a lack of statistical significance in the values observed.
- The same mistake was made in figure 5 and related text, regarding the expression of CCR2 in the intestine, there is no significant increase! Consider that the inflammation in the model may not be uniform, as pointed out earlier.
Conclusion:
There is no consistent conclusion, as the data in the way they were presented did not allow this. It is suggested to restructure the paper with the answers to the questions.
Author Response
Reviewer 1
Introduction: Most of the cited papers are more than 10 years old, we suggest updating these references and focusing more on the role of phagocytic cells in microsporidiosis.
Response: We have updated some new references, such as reference “Advancement in our understanding of immune response against Encephalitozoon infection” published in 2021 (DOI: 10.1111/pim.12828) and “Innate and Adaptive Immune Responses Against Microsporidia Infection in Mammals” published in 2020 (DOI: 10.3389/fmicb.2020.01468); as well as the ones about the roles of phagocytic cells, such as “Encephalitozoon cuniculi takes advantage of efferocytosis to evade the immune response” (DOI: 10.1371/journal.pone.0247658), in the introduction section.
Material and methods: Some concerns involve the experimental model and should be clarified:
- What is the inoculation route of E. hellem spores used for the infection?
Response: We used the intraperitoneal injection for E. hellem spores’ inoculation. We have updated this information now in “2.3 Experimental Design” of materials and methods.
- How was the infection diagnosed? Considering that not all animals become infected, as described in the base article for this model, perhaps some animals included in the infected group could not be infected? How to prove encephalitozoonosis?
Response: The infection was diagnosed by PCR assay. The whole genome of the intestine sections from either the E. hellem-infected or un-infected mice were extracted by CTAB method. The primers of E. hellem which result in a 279 bp target band were used to amplify and determine the presence of E. hellem. We have added this new figure as Figure 1A in the revised manuscript.
- How many days after infection were the collections performed?
Response: The collections performed at the 17th day after infection, mentioned in “2.3 Experimental Design”.
- Why use the intestine to identify monocytes?
Response: Because the intestine is prone to be infected by microsporidia which can be supported by the reference entitled Microsporidia-Host Interactions (doi: 10.1016/j.mib.2015.03.006), which was also included in our references as well now.
- Which tissues should be most affected by the infection?
Response: Intestinal tract should be most affected, as we mentioned above.
- What is the dose of dexamethasone used? It wasn't clear.
Response: The exact dose is 5 mg/kg for each mouse, added into the Method section -“2.3”.
To be more precise, monocyte and phagocyte populations should be identified in the gut, as done in the bone marrow. When inferring an inflammatory process in the intestine, it must be identified either by counting cells by cytometry or by histopathological analysis. Which infestinal segment was used for extraction? what is the extraction method? what weight of intestine did you use? The relative expression was not clarified! Considering that the expression refers to the number of cells is imprecise, since few cells can express many of the investigated molecules, does not seem to be a sufficient approach to answer the proposed questions.
Response: Thank you for the suggestion. In fact, we have measured the weight of the tissue for better assessment. About 0.32 g of intestine tissue was used each time, and the total RNA was extracted by TRIzol method. The specific information has been added in the manuscript.
Results:
- It is suggested that figures 1 and 2 are unified.
- The text referring to figure 4 suggests an increase in Ly6C in the intestine of infected mice, but the figure shows a lack of statistical significance in the values observed.
Response: Thanks for the comment. The expression level of Ly-6C did upregulate without the significance in infected mice. We have emphasized again in the revised manuscript. We believe that result is consistent with our hypothesis that the chronic infection of E. hellem is not that severe and cannot lead to severe immune response based on our work and previous studies. We have done related research about E. hellem could not evoke severe immune response, and the manuscript is about to publish.
- The same mistake was made in figure 5 and related text, regarding the expression of CCR2 in the intestine, there is no significant increase! Consider that the inflammation in the model may not be uniform, as pointed out earlier.
Response: We believe for the same reason mentioned above that E. hellem infection could not evoke severe immune response. We did prove by experiment that the expression level of CCR2 did upregulate without the significance in infected mice in our experiment. We have emphasized again in the revised manuscript.
Conclusion:
There is no consistent conclusion, as the data in the way they were presented did not allow this. It is suggested to restructure the paper with the answers to the questions.
Response: We have updated the conclusion part to better summarize our findings and already adjusted the paper according to your suggestions.
We sincerely thank for your close review and detailed points, and exactly help us to improve this manuscript.

Reviewer 2 Report
The presented article is at a very good level both scientifically and in the processing of the results, the comprehensibility and reproducibility is good. The reader will learn everything essential from the experiment, the experimental design of which was designed in an understandable and concise manner.
The presented article is at a very good level both scientifically and in the processing of the results, the comprehensibility and reproducibility is good. The reader will learn everything essential from the experiment, the experimental design of which was designed in an understandable and concise manner.
The authors sufficiently evaluated monocytes or macrophages as the most important immune cells, participating in the management of E. hellem infection in the first stage of the host's natural defense. Using appropriate modern methods, they achieved statistically significant results of relevant groups of monocytes and macrophages in the peripheral blood and in the small intestine during microsporidian infection, thereby contributing to the clarification of the host-pathogen interaction.
I recommend publishing the article in your journal.
Author Response
Thank you so much for your comments and support.

Reviewer 3 Report
Encephalitozoon hellem Infection Promotes Monocytes Extravasation.
L13, pl report the whole species name.
L 79 80 at the dose of 100ul/ each mouse daily persisted for six days.
Is the unit correct? Pl also keep space between the number and the unit through the whole ms.
Material and methods
All parts in m and m should be reported in detail and cite references from the methods used.
In the legends, the species' names must be written in cursive.
Linguistic improvements would be made in advance for the whole MS; pl ask Dr Timothy to check it again.
Example:
During the acute inflammatory phase of infection, Listeria monocytogenes, the monocytic left shift occurs in the bone marrow at about 72 hours [25]. The maturation and migration of dendritic cells are also reported to be affected by microsporidia infection [26]. Therefore, we are very interested in future studies to investigate in depth with a broader time the frame of microsporidia infection and the influences on more kinds of immune cells. These may lead to a more systematic understanding of innate immune response against microsporidia infection.
References
Pl, check all the refs again. The species names must be in cursive.
Author Response
Reviewer 3
Comments and Suggestions for Authors ‘Encephalitozoon hellem Infection Promotes Monocytes Extravasation.’
-L13, pl report the whole species name.
Response: The whole species name is Encephalitozoon hellem, we have added into the abstract.
-L 79 80 at the dose of 100ul/ each mouse daily persisted for six days.
Is the unit correct? Pl also keep space between the number and the unit through the whole ms.
Response: The exact dose for injection is 5 mg/kg for each mouse on a daily basis, we have corrected it in the article and added the space through the whole manuscript.
-Material and methods: All parts in m and m should be reported in detail and cite references from the methods used.
Response: We have added the references from the methods used in the part of materials and methods, such as the new added references 22 and 23.
-In the legends, the species' names must be written in cursive.
Response: We have corrected the form of species’ names-writing in the manuscript.
-Linguistic improvements would be made in advance for the whole MS; pl ask Dr Timothy to check it again.
Response: We have invited Dr. Keiffer to double check our language again. Thank you for the comment.
Example: (L256)
During the acute inflammatory phase of infection, Listeria monocytogenes, the monocytic left shift occurs in the bone marrow at about 72 hours [25]. The maturation and migration of dendritic cells are also reported to be affected by microsporidia infection [26]. Therefore, we are very interested in future studies to investigate in depth with a broader time the frame of microsporidia infection and the influences on more kinds of immune cells. These may lead to a more systematic understanding of innate immune response against microsporidia infection.
Response: We have adjusted the language accordingly.
-References
Pl, check all the refs again. The species names must be in cursive.
Response: We have already corrected the species names in cursive in the references.
Thank you so much for your detailed comments and your support.

Round 2
Reviewer 1 Report
All suggestions made were accepted and improved the understanding of the study. Just an open question, the position of the conclusions which was put as item 4, before the discussion. Authors need to review.
Author Response
-All suggestions made were accepted and improved the understanding of the study. Just an open question, the position of the conclusions which was put as item 4, before the discussion. Authors need to review.
Response: The position of the conclusion has been changed to the part 5, behind the discussion. Thank you for your suggestion.

Reviewer 3 Report
89-91
All the mice scarified on the 17th day after infection, and samples of bone marrow (from hind legs), peripheral blood, about o.32 g small intestines were collected respectively from each group.
sacrified !
229-230
host [U1]monocytes in the bone marrow proliferated in response to infection. The extravasation of [U2]monocytes also increased significantly upon reaching the site of
keep space after (U1)
Not all of my comments have been answered.
The details for the M and M are not answered in details
What about the language?
Author Response
-L89-91
All the mice scarified on the 17th day after infection, and samples of bone marrow (from hind legs), peripheral blood, about o.32 g small intestines were collected respectively from each group.
Sacrificed!
Response: We have changed the spelling of sacrificed.
-L229-230
host [U1]monocytes in the bone marrow proliferated in response to infection. The extravasation of [U2]monocytes also increased significantly upon reaching the site of keep space after (U1)
Response: We have added the space in the sentences of conclusion.
-Not all of my comments have been answered.
Response: Thank you for your comments, and we have addressed all.
They are as follows:
-L13, pl report the whole species name.
Response: The whole species name is Encephalitozoon hellem, we have added into the abstract.
-L 79 80 at the dose of 100ul/ each mouse daily persisted for six days.
Is the unit correct? Pl also keep space between the number and the unit through the whole ms.
Response: The unit written before was not correct. It has been changed to the injection dose: 5 mg/kg for each mouse on a daily basis, we have corrected it in the article and added the space through the whole manuscript.
-In the legends, the species' names must be written in cursive.
Response: We have corrected the form of species’ names-writing in the manuscript.
-References
Pl, check all the refs again. The species names must be in cursive.
Response: We have already corrected the species names in cursive in the reference 9 10 18 19 21 28 29 30 31 32.
-The details for the M and M are not answered in details
Response:
In the part of “2.2 pathogen”, the method of spore collection was cited from reference 22.
In the part of “2.3 Experimental Design”, the use of dexamethasone was cited from reference 21 and 23. The method of samples treatment were added as follows: TRIzol method was used to extract the total RNA of samples. CTAB method was used for the extraction of the whole genome in tissues. PCR assay was used to prove the infection.
In the part of “2.4 Bone marrow isolation”, some of experimental processes were cited by reference 24.
The methods used in other parts (2.5-2.8) were already specific and some of them are the experience-accumulating of our research group.
-What about the language?
Response: We have invited Dr. Keiffer to double check our language again, some descriptions were improved specifically.
And for the example given, we adjusted it to: In the acute inflammatory phase of Listeria monocytogenes infection, monocytic left shift occurs in the bone marrow at about 72 hours [31]. Such left shift was also reported in the chronic infection of E. hellem, which needs further exploration. Interestingly, the maturation and migration of dendritic cells are reported to be inhibited by microsporidia infection [32]. Therefore, we are very interested in future studies to investigate in depth that how microsporidia infection influences dendritic cells and some other kinds of innate immune cells.
Thank you sincerely for the comments and your support.
